# Potential Distribution and Identification of Critical Areas for the Preservation and Recovery of Three Species of *Cinchona* L. (Rubiaceae) in Northeastern Peru

Elver Coronel-Castro [1,2,*] , Gerson Meza-Mori [1] , Jose M. Camarena Torres [1], Elí Pariente Mondragón [1,2,3] , Alexander Cotrina-Sanchez [4] , Manuel Oliva Cruz [1] , Rolando Salas López [1] and Ricardo E. Campo Ramos [1,5,*]

1 Instituto de Investigación para el Desarrollo Sustentable de Ceja de Selva (INDES-CES), Universidad Nacional Toribio Rodríguez de Mendoza de Amazonas (UNTRM), Chachapoyas 01001, Peru; gmeza@indes-ces.edu.pe (G.M.-M.); 4658844572@untrm.edu.pe (J.M.C.T.); eli.pariente@untrm.edu.pe (E.P.M.); manuel.oliva@untrm.edu.pe (M.O.C.); rolando.salas@untrm.edu.pe (R.S.L.)

2 Herbario KUELAP, Facultad de Ciencias Agrarias, Universidad Nacional Toribio Rodríguez de Mendoza de Amazonas (UNTRM), Chachapoyas 01001, Peru

3 Instituto de Investigación, Innovación y Desarrollo para el Sector Agrario y Agroindustrial de la Región Amazonas (IIDAA), Universidad Nacional Toribio Rodríguez de Mendoza de Amazonas (UNTRM), Chachapoyas 01001, Peru

4 Department for Innovation in Biological, Agri-Food and Forest Systems, Università degli Studi della Tus-cia, Via San Camillo de Lellis 4, 01100 Viterbo, Italy; alexander.cotrina@unitus.it

5 Facultad de Ingeniería Civil y Ambiental, Universidad Nacional Toribio Rodríguez de Mendoza de Amazonas (UNTRM), Chachapoyas 01001, Peru

* Correspondence: elver.coronel@untrm.edu.pe (E.C.-C.); ricardo.campos@untrm.edu.pe (R.E.C.R.); Tel.: +51-941957122 (E.C.-C.); +51-986281544 (R.E.C.R.)

**Abstract:** The genus *Cinchona* L. has important medicinal, cultural, and economic value and is the emblematic tree of Peru. The genus is mainly found in the cloud forests of the Andes. However, the expansion of agriculture and livestock farming in the department of Amazonas is degrading these ecosystems and has reduced the size of the genus's populations. In this work, we model the potential distribution under current conditions of three *Cinchona* species (*C. capuli* L. Anderson, *C. macrocalyx* Pav. Ex DC., and *C. pubescens* Vahl.) to identify areas with a high likelihood of species presence and their key conservation and reforestation zones. We fitted a maximum entropy (MaxEnt) model using nineteen bioclimatic variables, three topographic variables, nine edaphic variables, and solar radiation. Under current conditions, the potential distribution of *C. capuli* covers 17.22% (7243.98 km$^2$); *C. macrocalyx*, 29.11% (12,238.91 km$^2$); and *C. pubescens*, 22.94% (9647.63 km$^2$) of the study area, which was mostly located in central and southern Amazonas. Only 24.29% (25.51% of *C. capuli*, 21.02% of *C. macrocalyx,* and 26.35% of *C. pubescens*) of the potential distributions are within protected areas, while 10,987.22 km$^2$ of the surface area of the department of Amazonas is degraded, of which 29.80% covers the area of probable occurrence of *C. capuli*, 38.72% of *C. macrocalyx*, and 34.82% of *C. pubescens*. Consequently, it is necessary to promote additional conservation strategies for *Cinchona*, including the establishment of new protected areas and the recovery of degraded habitats, in order to protect this species.

**Keywords:** *Cinchona*; potential distribution; Maxent; restoration; quina tree; Amazonas

## 1. Introduction

Andean montane forests hold globally significant levels of biodiversity, harboring an exceptional variety of species and much endemism [1,2]. These ecosystems provide essential ecosystem services, such as carbon sequestration [3] and streamflow regulation during dry seasons [4]. In addition, the organic soil of these forests has unique hydraulic properties, including high water infiltration and retention [5,6]. However, their steep topography makes them vulnerable to erosion during heavy rains [7]. Although they face significant

threats such as climate change and natural disturbances [8,9], it is human activities (population expansion and resource extraction) that are accelerating their deterioration [7]. These pressures are growing and are expected to continue due to the persistent demand for forest resources and food [10].

The *Cinchona* genus originates from the Andean valleys of South America [11] and is located in the tropical region of the Andes mountain range [12,13]. The greatest diversity of species is observed in the mountains of Peru [14], with northern Peru identified as one of the genus's diversity hotspots [15]. Around 12 species of *Cinchona* have been documented in the Amazonas department, mainly inhabiting mountain forests between 600 and 3400 m above sea level. However, it faces threats from human activities such as agriculture, livestock farming, and urban expansion [16] that negatively impact *Cinchona* populations [13,15–17]. Research efforts in the Amazonas department have predominantly focused on aspects such as the diversity and taxonomy of the genus [18], as well as its propagation [16,17,19]. Recent studies on *Cinchona* in Peruvian territory and the Amazonas department have analyzed its distribution as a focus for conservation and restoration [20–22]. Despite these efforts, there is a lack of more-detailed investigations on how different species of *Cinchona* are distributed. These species have different habitats, environmental conditions, uses, and conservation levels, requiring more specific studies to better understand population dynamics and identify effective conservation strategies to implement.

Species distribution models (SDMs) are tools that link species occurrence information to a range of bioclimatic, topographic, and edaphic variables. These tools play a vital role in protecting species, studying biogeography, and obtaining information on the impacts of climate change [23–26]. Within the field of SDMs, the maximum entropy algorithm (MaxEnt) estimates the potential distribution of a species, based on the principle that the most accurate prediction is achieved by optimizing the entropy of the distribution under specific environmental conditions [26]. This model stands out for its superior predictive accuracy compared to other models and its ability to work efficiently with small sample sizes [27,28]. The MaxEnt algorithm has been used extensively in modelling the distribution of various species, addressing present conditions and climate change scenarios [28]. It has been applied in a wide range of contexts, such as plant conservation, especially for endangered and endemic species [29–34]. It has also been useful in modeling the distribution of invasive species, forestry management, and agriculture [35–38].

In this work, the potential distribution of three *Cinchona* species (*C. capuli*, *C. macrocalyx*, and *C. pubescens*) in the department of Amazonas, northeastern Peru, was modeled. The research delved into how these species are connected to conservation areas and how regional degradation could impact them. Given their status as forest species, they face several threats due to their timber value and preference for fertile soils, often converted for agricultural and livestock purposes. The aim of this analysis extended beyond merely mapping the current distribution of these species; it also sought to identify protected areas and those with potential for their conservation and restoration, acknowledging their endangered status. To address this, specific questions were posed: (1) What is the potential distribution area of *C. capuli*, *C. macrocalyx*, and *C. pubescens* in the department of Amazonas? (2) Do the potential distribution areas of these species align with the conservation areas within the department? (3) Is it necessary to conserve and restore the distribution area of these *Cinchona* species? For each species, (i) a baseline file of georeferenced presence records and environmental variables was constructed, (ii) maps of currently suitable environmental habitats were generated, and (iii) potential areas for conservation and restoration were identified.

## 2. Materials and Methods

### 2.1. Study Area

This research was carried out in the department of Amazonas, located in the northeast of the Peruvian Andean zone (Figure 1). Amazonas covers about 42,050 km$^2$ of mountainous area, mostly covered by the closed canopy rainforest and Andean Forest, with

geographical coordinates ranging from 3°0′ to 7°2′ south latitude and 77°0′ to 78°42′ west longitude. This area exhibits a significant variation in altitude, ranging from 120 m above sea level in the north to 4900 in the south. The department stands out for its impressive ecological diversity, manifested in the presence of four distinct ecosystems: lowland forest, highland forest or yunga, the Andean forests and grasslands, and tropical dry forest [39].

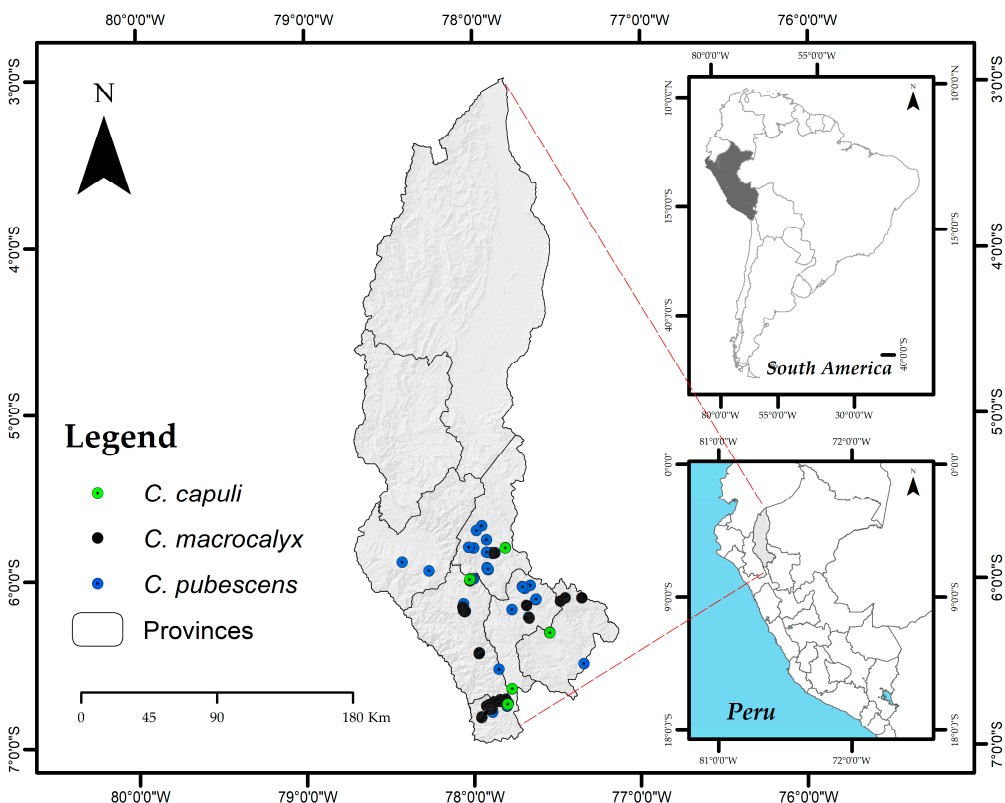

**Figure 1.** Occurrence data of *Cinchona capuli*, *C. macrocalyx*, and *C. pubescens* in Amazonas, northeastern Peru.

The department of Amazonas exhibits a wide range of climatic conditions, ranging from hot and humid areas to hot and dry regions. Temperatures vary significantly, with highs of 40 °C in the northern lowland forests and lows of 2 °C in the southern highlands. Water availability is also variable in the department, with some areas experiencing an annual shortage of around 924 mm of precipitation, while other areas receive an excess of up to 3000 mm per year [39]. Amazonas is the department in Peru with the third-highest number of conservation areas, including natural protected areas (NPA), regional conservation areas (RCA), and private conservation areas (PCA) [40]. Despite these conservation measures, approximately 11,000 km$^2$ (27.4%) of the department's territory has suffered degradation [41]. This is due to inappropriate agricultural practices [42–44], unsustainable forest use [45], deforestation driven by road construction, and the unregulated expansion of urban areas [46–48].

### 2.2. Records of Cinchona Species

Records of *Cinchona capuli*, *C. macrocalyx*, and *C. pubescens* (Figure S1) in the department of Amazonas, northeastern Peru, were collected via field trips conducted between March 2021 and June 2023. These trips were made to collect botanical samples of these genera. Simultaneously, specimens were stored in the KUELAP, USM, and MOL herbaria were examined. Photographs of exsiccata from international herbaria, such as ECON, K, MO, and US, were also collected [49]. Additionally, occurrence data previously selected from the Global Biodiversity Information Facility (GBIF) (https://www.gbif.org/ (accessed on 2 April 2023)) were utilized following a taxonomic review of the samples. The records that

lacked a photograph of the specimen or presented uncertain data were excluded. In total, 110 records of the three *Cinchona* species were obtained for the department, including 16 on *C. capuli*, 33 on *C. macrocalyx*, and 61 on *C. pubescens* (Figure 1, Table S1).

### 2.3. Bioclimatic, Environmental, Topographic, and Edaphic Variables

A literature review was conducted to identify variables that could contribute to the distribution models [50]. This included 19 bioclimatic variables and one environmental variable (radiation), obtained from the WorldClim version 2 database at 30 s (~1 km) resolution (http://www.worldclim.org/ (accessed on 22 June 2023)) [51]; three topographic variables (elevation, aspect, and slope) derived from the Digital Elevation Model (DEM) at a spatial resolution of 250 m downloaded from the CGIAR Consortium for Spatial Information portal (http://srtm.csi.cgiar.org/ (accessed on 26 June 2023)) [52]; and nine edaphic variables, obtained from Soil Grids (https://soilgrids.org/ (accessed on 29 June 2023)) at a spatial resolution of 250 m. The bioclimatic and environmental layers were used in their current conditions (1970–2000 average), as they are widely used in ecological studies due to their free availability, global coverage, and good quality [53]. The data sets were re-sampled at a resolution of 250 m, and in total, 32 thematic layers were obtained. All variables were converted to the ASCII (American Standard Code for Information Interchange) format (Table 1).

**Table 1.** Variables chosen for the distribution model.

| Variable | Symbol | Units |
| --- | --- | --- |
| 1. Environmental variables | | |
| 1.1. Bioclimatic | | |
| Annual Mean Temperature | bio01 | °C |
| Mean Diurnal Range | bio02 | °C |
| Isothermality | bio03 | °C |
| Temperature Seasonality | bio04 | °C |
| Max Temperature of Warmest Month | bio05 | °C |
| Min Temperature of Warmest Month | bio06 | °C |
| Annual Temperature Range | bio07 | °C |
| Mean Temperature of Wettest Quarter | bio08 | °C |
| Mean Temperature of Driest Quarter | bio09 | °C |
| Mean Temperature of Warmest Quarter | bio10 | °C |
| Mean Temperature of Coldest Quarter | bio11 | °C |
| Annual Precipitation | bio12 | Millimeter |
| Precipitation of Wettest Month | bio13 | Millimeter |
| Precipitation of Driest Month | bio14 | Millimeter |
| Precipitation Seasonality | bio15 | Millimeter |
| Precipitation of Wettest Quarter | bio16 | Millimeter |
| Precipitation of Driest Quarter | bio17 | Millimeter |
| Precipitation of Warmest Quarter | bio18 | Millimeter |
| Precipitation of Coldest Quarter | bio19 | Millimeter |
| 1.2. Radiation | radiac | $MJ\ m^{-2}\ día^{-1}$ |
| 2. Topographies | | |
| Elevation above mean sea level | dem | mals |
| Cardinal orientation of the slope | aspect | ° |
| Slope of the terrain | slope | ° |
| 3. Edaphic variables | | |
| Bulk density of the fine earth fraction | bdod | $cg/cm^3$ |
| Clay content | clay | % |
| Volumetric fraction of coarse fragments | coarse | $cm^3/dm^3$ (vol %) |
| Sand content | sand | % |
| Slime content | Silt | % |
| Cation exchange capacity | cec | $cmol\ kg^{-1}$ |
| Total nitrogen | nitrog | cg/kg |
| Organic carbon | soc | $g\ kg^{-1}$ |
| pH in $H_2O$ | pH | pH × 10 |

### 2.4. Variable Selection

The variables used in the distribution models may exhibit a high degree of correlation, potentially leading to less-reliable predictions due to the similar contributions of two or more variables [27,53]. To address this issue and mitigate multicollinearity [54,55], the following criteria were considered: (i) extracting pixel values of the 32 variables from georeferenced species records [56–58]; (ii) determining the number of variables groups using K-means clustering with the factoextra package in R (Figure 2), which served as input for (iii) constructing dendrograms and selecting one variable per group [54] (Figure 3). However, in cases where certain groups contained more than one variable, a Pearson correlation analysis was conducted within each group, and variables with higher correlation ($|r| > 0.7$) were removed. This threshold is considered appropriate to prevent collinearity from distorting model estimation [54,59–61]. The application of these criteria resulted in the selection of a subset of eight variables for *C. capuli* (Figure 2a) and *C. macrocalyx* (Figure 2b), and nine for *C. pubescens* (Figure 2c).

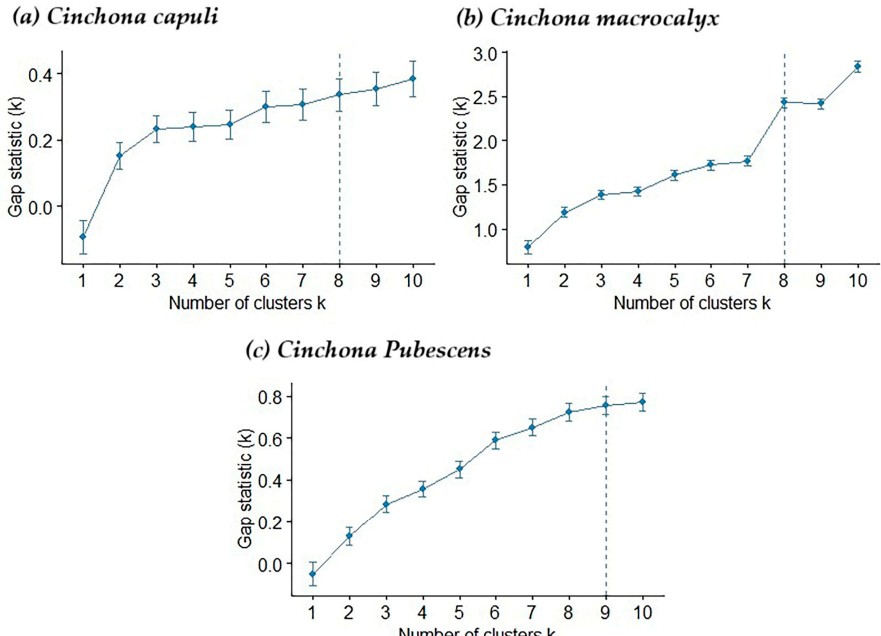

**Figure 2.** Optimal number of clusters among bioclimatic variables for modeling the potential distribution of (**a**) *Cinchona capuli,* (**b**) *C. macrocalyx* y (**c**) *C. pubescens* in the department of Amazonas, northeastern Peru.

### 2.5. Model Construction

The maximum entropy algorithm was used to create potential distribution models, utilizing the open-source software MaxEnt version 3.4.1. The georeferenced records were randomly divided into two sets: one for training (75% of the records) and one for validation (25% of the records) of each model. Ten replicates were made for each species using the bootstrap method with a maximum of 1000 iterations each, enabling the algorithm to make a more realistic prediction [26]. The convergence threshold was set at 0.00001, meaning that the algorithm continued to iterate until the difference between successive iterations was lower than this value. Additionally, a maximum limit of 10,000 background points was set for the modelling process [62]. The other default settings were left unchanged, due to MaxEnt's ability to automatically select the most appropriate function based on the amount of data available for the model. This automatic selection of settings is important to optimize the performance of the model by adapting it to the specific characteristics of the available data [28]. To validate the models, the area under the curve (AUC) method, derived from the receiver operating characteristic (ROC) [63], was used. This method assigns a score that reflects the model's predictive performance. Based on the AUC values obtained, five

differentiated performance levels were established [64]: "excellent" for AUC > 0.9, "good" for AUC in the range of 0.8–0.9, "acceptable" for AUC between 0.7 and 0.8, "poor" for AUC in the range of 0.6–0.7, and "invalid" for AUC less than 0.6. The main advantage of this approach is its independence from subjective thresholds, which gives a higher degree of objectivity in assessing results [65]. The results were presented in a logistic format [66], generating a map of continuous probability values ranging from 0 to 1 for the potential distribution ranges. These values were then categorized into four potential distributions [31,32,66]: "high" for values greater than 0.6, "moderate" for values between 0.4 and 0.6, "low" for values between 0.2 and 0.4, and "no potential distribution" for values below 0.2.

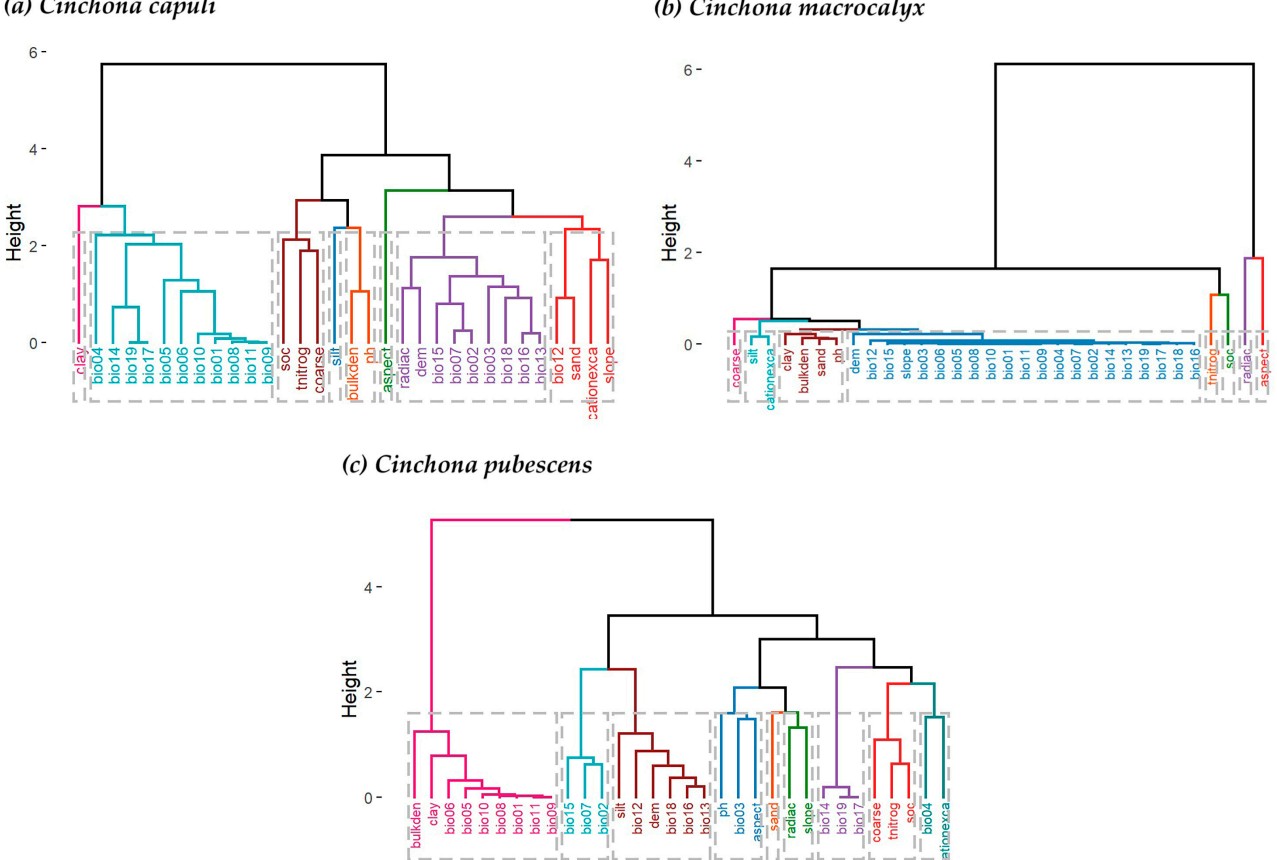

**Figure 3.** Dendrograms illustrating the clusters analysis of bioclimatic variables for modeling the potential distribution of *C. capuli* (**a**), *C. macrocalyx* (**b**), and *C. pubescens* (**c**) in Amazonas, northeastern Peru.

### 2.6. Conservation and Recovery of Cinchona Species

A superimposition process was performed using the three maps, each depicting the current potential distribution of one of the three *Cinchona* species. This process utilized the Shapefile from the Peru Protected Natural Areas System, last updated in 2023 and accessible on the SERNANP geoserver (available at https://geo.sernanp.gob.pe/visorsernanp/# (accessed on 11 September 2023)). The aim was to identify areas within the department where protected *Cinchona* species are located under various conservation modalities. These modalities include national parks (NPs), national sanctuaries (NSs), communal reserves (CRs), reserved zones (RZs), buffer zones (BZs), regional conservation areas (RCAs), and private conservation areas (PCAs) [40]. Additionally, the layers displaying the potential distributions of *Cinchona* species were overlaid onto the map of degraded areas in Peru (2001–2021). This map, crafted by the Ministry of the Environment, was based on estimating the total or partial loss of certain essential ecosystem components such as water, soil,

and species. This loss impacts the natural structure and functioning of these ecosystems, diminishing their ability to sustain diverse living organisms, including humans; that is, their capacity to provide ecosystem services. Indicators recommended by the United Nations Convention to Combat Desertification and Drought on Land Degradation Neutrality (LDN) were utilized [41]. This analysis allowed the identification of areas within the department where *Cinchona* species have distribution potential but are currently degraded, showing potential for restoration. Additionally, it pinpointed zones of potential species distribution within conservation areas or those experiencing no degradation, which could be suitable for conservation efforts [40,41].

## 3. Results

### 3.1. Contribution of the Variables

The contributions of environmental variables (Table S1) and the results of the jackknife test (Figure S2) revealed the cumulative contribution of classified variables affecting the habitat suitability of the three *Cinchona* species as follows: edaphic variables (80.3% for *C. capuli*, 79.4% for *C. macrocalyx*, and 13.1% for *C. pubescens*), climatic variables (19.5% for *C. capuli*, 2.2% for *C. macrocalyx*, and 86.3% for *C. pubescens*), and topographic variables (0.2% for *C. capuli*, 18.4% for *C. macrocalyx*, and 0.6% for *C. pubescens*). However, at a general level, it was observed that only three variables in each model significantly contributed. In the distribution model of *C. capuli*, the variables cec, bio03, and bio04 accounted for 99.7% of the contribution, whereas for *C. macrocalyx*, the variables cec, aspect, and clay contributed 94.7% to the distribution model. Additionally, the variables bio06, bio14, and cec contributed 91.1% in the distribution model of *C. pubescens* (Table 2).

**Table 2.** Variables with major contributions to MaxEnt modelling of the three *Cinchona* species in Amazonas, Peru.

| Specie | Variable 1 (%) | Variable 2 (%) | Variable 3 (%) | Total Contribution |
|---|---|---|---|---|
| *C. capuli* | cec (80.2%) | bio03 (13.9%) | bio04 (5.6%) | 99.7% |
| *C. macrocalyx* | cec (69.2%) | aspect (18.4%) | clay (7.1%) | 94.7% |
| *C. pubescens* | bio06 (62.4%) | bio14 (20.6%) | cec (8.1%) | 91.1% |

### 3.2. Performance of the Distribution Pattern of the Model

Three current distribution models were generated for the Cinchona species. The results showed that all three species achieved area under the curve (AUC) values greater than 0.9, indicating excellent predictive performance in each case (as detailed in Table 3). These results demonstrate a highly reliable capability of predicting the distribution of these species (Figure S2).

**Table 3.** Performance of the MaxEnt model of the three *Cinchona* species in the department of Amazonas.

| Specie | *C. capuli* | *C. macrocalyx* | *C. pubescens* |
|---|---|---|---|
| AUC | 0.992 | 0.981 | 0.995 |

### 3.3. Occurrence of Cinchona Species

Under current edaphoclimatic conditions, the potential distribution (high, moderate, and low) of the three *Cinchona* species in the department of Amazonas was determined. *C. capuli* covers an area of 7243.98 km$^2$ (1891.76 km$^2$ high, 2171.41 km$^2$ moderate, and 3180.81 km$^2$ low probability), representing 17.22% of the department, *C. macrocalyx* has a potential distribution area of 12 238.91 km$^2$ (5120.35 km$^2$ high, 3073.51 km$^2$ moderate, and 4045.05 km$^2$ low probability), accounting for 29.11% of the study area. *C. pubescens* covers an area of 9647.63 km$^2$ (4142.04 km$^2$ high, 2348.38 km$^2$ moderate, and 3157.21 km$^2$ low probability), equivalent to 22.94% of the territory of Amazonas (Figure 4). The distribution

of these species is mainly concentrated in the central and southern parts of the department in the provinces of Bongará, Chachapoyas, Luya, and Rodríguez de Mendoza. In the north, the province of Utcubamba also shows potential for the distribution of these species, particulary *C. mcacrocalyx*. Mountain ecosystems, such as the montane and altimontane forests (Pluvial) of Yunga, present a greater distribution potential for these species. However, *C. macrocalyx* can also be found in Jalca and grasslands (Figure 5).

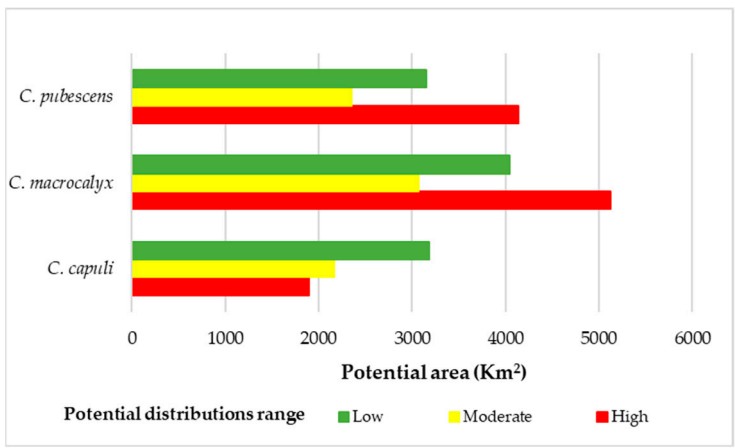

**Figure 4.** Potential distribution areas of *C. capuli*, *C. macrocalyx*, and *C. pubescens* in the department of Amazonas, northeastern Peru.

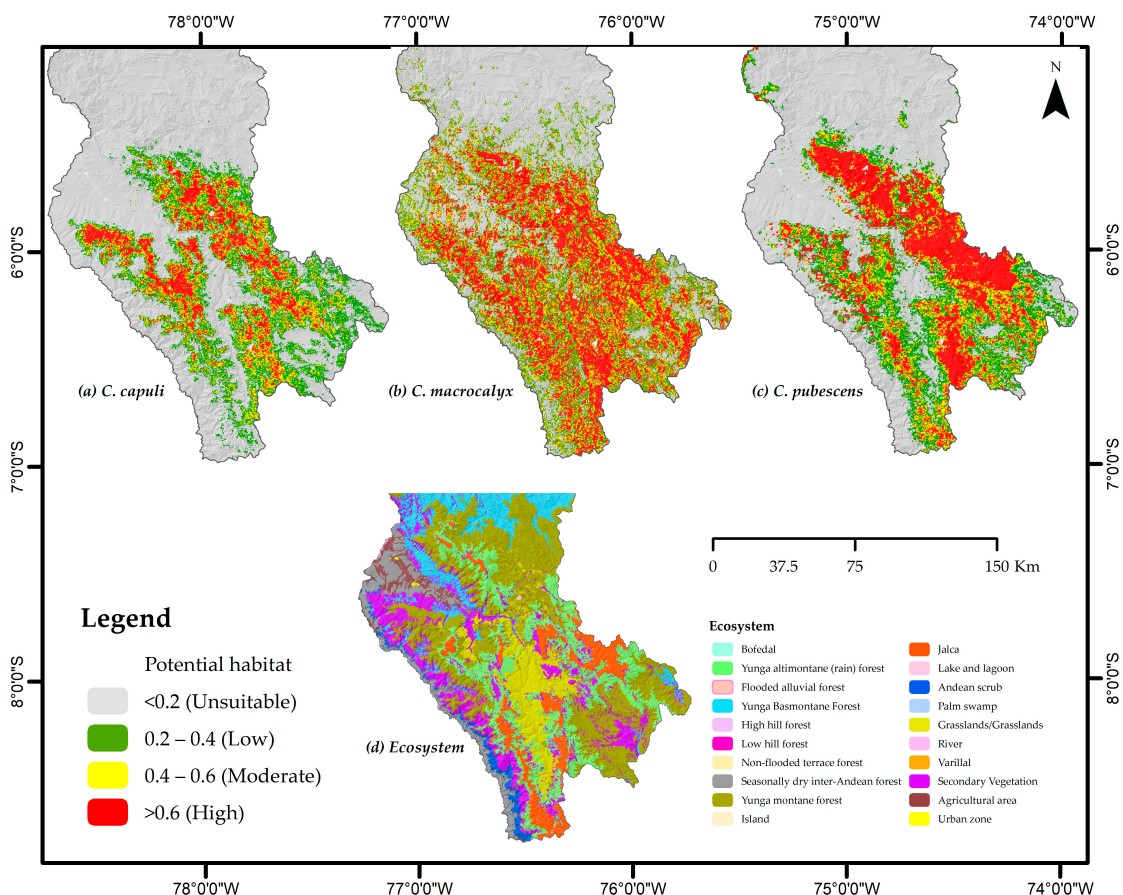

**Figure 5.** Potential distribution of three *Cinchona* species in the department of Amazonas, and their relationship with provinces and ecosystems. Where: (**a**) potential distribution of *C. capuli*, (**b**) potential distribution of *C. macrocalyx*, (**c**) potential distribution of *C. pubescens* and (**d**) ecosystems present in the department.

### 3.4. Conservation of Cinchona Species and Recovery Areas

In the department of Amazonas, it was found that 25.51% of the area with potential occurrence for *C. capuli* is located within conservation areas, covering a surface area of 1848.39 km$^2$. This represents 14. 98% of the total surface area of the conservation areas and includes 4.60% (113.92 km$^2$) of the territory of national sanctuaries (NSs), 39.06% (280.17 km$^2$) of regional conservation areas (RCAs), 67.96% (753.89 km$^2$) of private conservation areas (PCAs), 0.67% (9.15 km$^2$) of reserved zones (RZs), and 10.37% (691 km$^2$) of the buffer zone (BZ). In total, 370.84 km$^2$ of high-probability areas for *C. capuli* occurrence are within these conservation areas.

Similarly, 21.02% (2572.95 km$^2$) of the area with potential for *C. macrocalyx* occurrence is located within conservation areas, constituting 20.86% of the surface areas. This encompasses 10.12% (250.45 km$^2$) of NS, 48.03% (344.53 km$^2$) of RCA, 86.83% (963.15 km$^2$) of PCA, 6.02% (82.28 km$^2$) of RZ, and 13.99% (932.54 km$^2$) of BZ territories. The probability of *C. macrocalyx* occurrence is high in 8.38% (1033.98 km$^2$) of the conservation areas.

Finally, regarding *C. pubescens*, 26.34% (2541.86 km$^2$) of the area with potential occurrence lies within conservation areas, which is 20.60% of the total are. This includes 10.29% (254.67 km$^2$) of NS, 56.65% (406.37 km$^2$) of RCA, 86.21% (956.24 km$^2$) of PCA, 0.93% (12.7 km$^2$) of RZ, and 13.68% (911.88 km$^2$) of BZ. Finally, in the conservation areas of Amazonas, 10.15% are highly probable for the potential distribution of *C. pubescens* (Table 4).

**Table 4.** Potential distribution areas of *C. capuli*, *C. macrocalyx*, and *C. pubescens* that are protected in conservation areas present in the department of Amazonas, northeastern Peru.

| Specie | PNA Modalities | Geographic Area (km$^2$) | Potential Area (km$^2$) | | | | | | Total (km$^2$) | % |
|---|---|---|---|---|---|---|---|---|---|---|
| | | | Low | % | Moderate | % | High | % | | |
| *C. capuli* | NS | 2475.25 | 73.9 | 2.99 | 26.58 | 1.07 | 13.44 | 0.54 | 113.92 | 4.60 |
| | RCA | 717.35 | 228.39 | 31.84 | 47.68 | 6.65 | 4.1 | 0.57 | 280.17 | 39.06 |
| | PCA | 1109.25 | 268.73 | 24.23 | 263.12 | 23.72 | 222.09 | 20.02 | 753.89 | 67.96 |
| | RA | 1366.85 | 9.15 | 0.67 | 0 | 0.00 | 0 | 0.00 | 9.15 | 0.67 |
| | BZ | 6668.07 | 348.18 | 5.22 | 211.87 | 3.18 | 131.21 | 1.97 | 691.26 | 10.37 |
| | Total | *12,336.77* | *928.35* | *7.53* | *549.25* | *4.45* | *370.84* | *3.01* | 1848.39 | 14.98 |
| *C. macro-calyx* | NS | 2475.25 | 120.04 | 4.85 | 44.84 | 1.81 | 85.57 | 3.46 | 250.45 | 10.12 |
| | RCA | 717.35 | 164.18 | 22.89 | 91.26 | 12.72 | 89.09 | 12.42 | 344.53 | 48.03 |
| | PCA | 1109.25 | 235.56 | 21.24 | 259.18 | 23.37 | 471.41 | 42.50 | 963.15 | 86.83 |
| | RA | 1366.85 | 67.77 | 4.96 | 13.26 | 0.97 | 1.25 | 0.09 | 82.28 | 6.02 |
| | BZ | 6668.07 | 331.54 | 4.97 | 214.87 | 3.22 | 386.66 | 5.80 | 932.54 | 13.99 |
| | Total | *12,336.77* | 919.09 | 7.45 | 623.41 | 5.05 | 1033.98 | 8.38 | 2572.95 | 20.86 |
| *C. pubescens* | NS | 2475.25 | 75.48 | 3.05 | 38.54 | 1.56 | 140.65 | 5.68 | 254.67 | 10.29 |
| | RCA | 717.35 | 199.31 | 27.78 | 114.23 | 15.92 | 92.83 | 12.94 | 406.37 | 56.65 |
| | PCA | 1109.25 | 167.62 | 15.11 | 227.66 | 20.52 | 561.26 | 50.60 | 956.24 | 86.21 |
| | RA | 1366.85 | 9.62 | 0.70 | 2.05 | 0.15 | 1.03 | 0.08 | 12.7 | 0.93 |
| | BZ | 6668.07 | 161.85 | 2.43 | 153.56 | 2.30 | 596.47 | 8.95 | 911.88 | 13.68 |
| | Total | *12,336.77* | *613.88* | *4.98* | *536.04* | *4.35* | *1251.59* | *10.15* | 2541.86 | 20.60 |

The three *Cinchona* species are all distributed in one NS (Cordillera Colán National Sanctuary—CCNS), one RCA (except for *C. macrocalyx*, which is present two RCAs), seventeen PCAs, one RZ (Río Nieva), and the BZ of both the Alto Mayo Protected Forest and the CCNS (Figure 6).

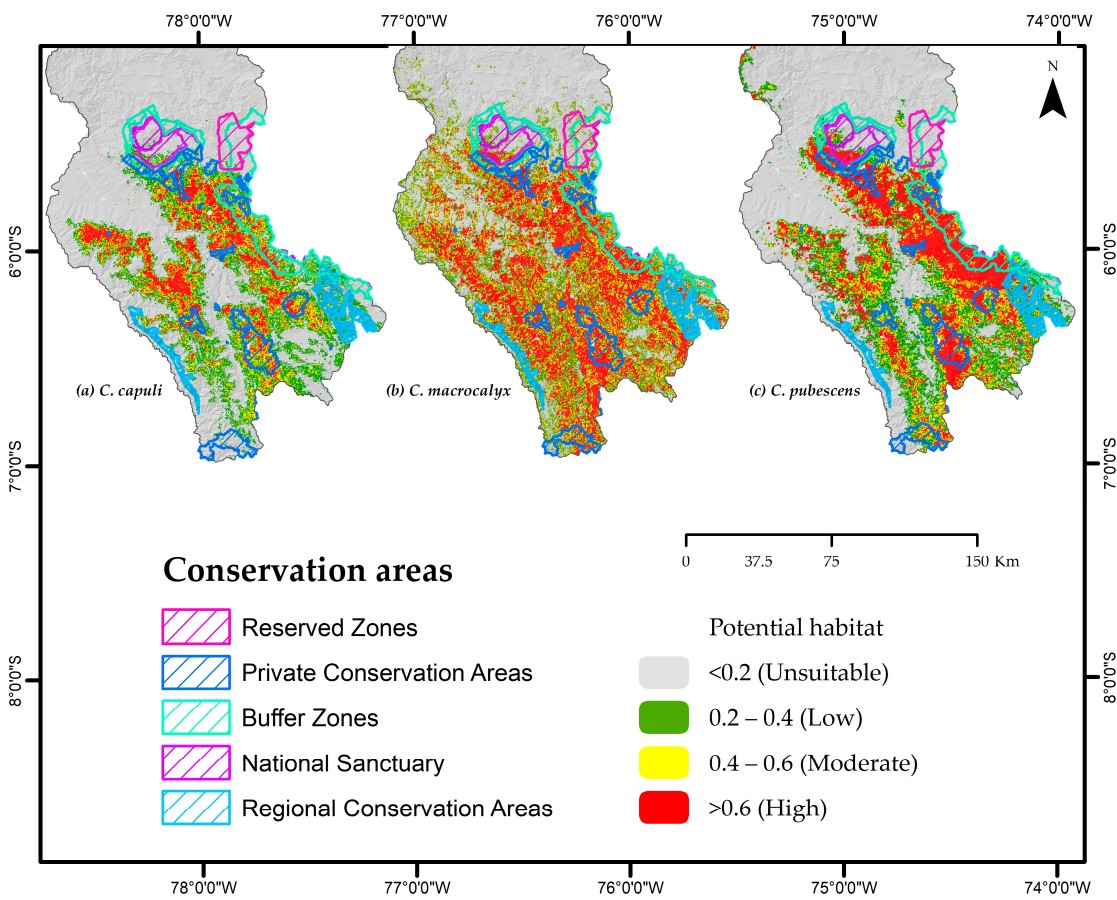

**Figure 6.** Relationship between potential distribution areas of *Cinchona* species and conservation areas (CA) in the department of Amazonas. Where: (**a**) potential distribution of *C. capuli* in CA, (**b**) potential distribution of *C. macrocalyx* in CA and (**c**) potential distribution of *C. pubescens* in CA.

In the study area, it was identified that approximately 10,987.22 km$^2$ of the territory has experienced degradation. Of this, 29.80% overlaps with the zone of potential presence of *C. capuli*, 38.72% with that of *C. macrocalyx*, and 34.82% with the zone of *C. pubescens* (Table 5, Figure 7). Furthermore, within these degraded areas, 1018.42 km$^2$, 1962.38 km$^2$, and 1719.69 km$^2$ were found to have a high probability of harboring *C. capuli*, *C. macrocalyx*, and *C. pubescens*, respectively. Consequently, applying appropriate conservation and management measures could potentially recover these areas.

**Table 5.** Areas with recovery potential for conserving and recovering the three species of *Cinchona* in the department of Amazonas, Peru.

| Specie | Degraded Areas | Geographic Area (km$^2$) | Potential Area (km$^2$) | | | | | | Total (km$^2$) | % |
|---|---|---|---|---|---|---|---|---|---|---|
| | | | Low | % | Moderate | % | High | % | | |
| *C. capuli* | Department | | 1203.47 | 10.95 | 1052.31 | 9.58 | 1018.42 | 9.27 | 3274.2 | 29.80 |
| *C. macrocalyx* | of | 10,987.22 | 1232.15 | 11.21 | 1059.99 | 9.65 | 1962.38 | 17.86 | 4254.52 | 38.72 |
| *C. pubescens* | Amazonas | | 1118.52 | 10.18 | 987.98 | 8.99 | 1719.69 | 15.65 | 3826.19 | 34.82 |

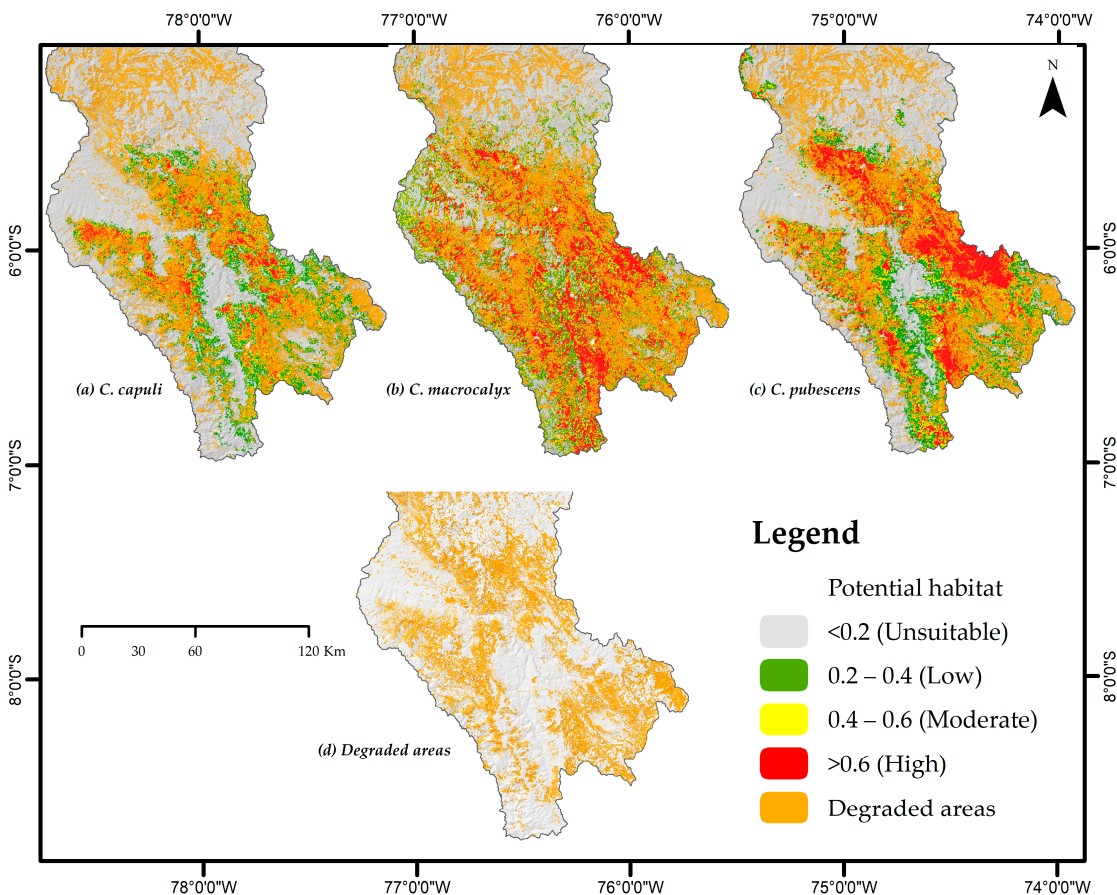

**Figure 7.** Potential distribution areas of (**a**) *C. capuli*, (**b**) *C. macrocalyx*, and (**c**) *C. pubescens* located within degraded areas in the department of Amazonas. Where: (**d**) represents the degraded area in the department.

## 4. Discussion

This paper presents the potential distribution of three species of the genus *Cinchona* (*C. capuli*, *C. macrocalyx*, and *C. pubescens*) in the department of Amazonas, northeastern Peru. The study specifically focused on their spatial distribution in conservation and degraded areas to identify habitats of these *Cinchona* species with potential for recovery in the department. This is the first attempt to use the species distribution model (SDM) with the maximum entropy modelling technique to predict and identify areas likely to harbor *C. capuli*, *C. macrocalyx*, and *C. pubescens* in Amazonas [67]. This model has previously been applied at departmental [21], regional [22,68], national, and global levels [20]. In contrast to prior studies on modelling the distribution of the genus [20–22,68], this study not only utilized occurrence data from virtual databases, herbarium databases, and floristic inventories, but also included species collected in the field. These species were taxonomically identified using morphological and morphometric characteristics [13–15]. Additionally, we thoroughly reviewed data related to botanical samples deposited in herbaria to build a database reflecting the actual presence of the studied species. This approach is crucial as the genus *Cinchona* can be confused with its close relatives *Remijia* and *Ladenbergia* [69]; approximately 330 names have been attributed to the *Cinchona* species, most of which are currently synonyms and/or correspond to related genera [11,14]. Furthermore, the biodiversity data obtained from GBIF often contain taxonomic and georeferencing errors [70], which can lead to incorrect model calibrations [71]. Hence, our study was based on a maximum entropy model and is thus believed to be better than previous studies in predictive accuracy and tolerance, especially considering the smaller scale of our sample (at the department level) [28]. The results of this analysis predict more specific distribution (at the species level,

as opposed to the entire genus, as in previous studies), enabling the establishment of more effective forest management strategies [31,72].

Spatial distribution plays a key role in understanding a species [72]. In this study, a model based on current data was used [70]. The analysis required several species-specific variables including nineteen bioclimatic, three topographic, and nine edaphic variables, as well as solar radiation variables [73]. It was determined that only 1891.76 km² of the Amazonas department's territory exhibits a high potential distribution area for *C. capuli*. This may be attributed to the species's limited occurrence in Peruvian territory, previously recorded only in the department of Piura, northern Peru [13,15], and in southern Ecuador [14,74]. *C. macrocalyx* and *C. pubescens* are species with larger areas of high probability of occurrence in Amazonas, comprising 5120.35 km² and 4142.04 km², respectively. The potential distribution of these species is supported by the fact that *C. macrocalyx* is mainly found in the Loja macro-region and specifically in the southern and northern Andean areas of the south of Ecuador, extreme north of Peru, central southern Peru, and Bolivia. *C. pubescens* is found mainly along the slope of the South American Andes [74]. A high probability of encountering these three species can be primarily expected in the central and southern parts of the Amazonas department, a prediction that coincides with Vergara et al. [22], who identified this department as having the potential for the occurrence for the entire *Cinchona* genus in Peru. Another reason for the possible distribution potential of the species in this area is the prevalence of cloud forests (montane forests), which are suitable habitats for the genus [11,75]. Contrarily, García et al. [20] did not identify Amazonas as a department with a high likelihood of potential distribution areas for *Cinchona*, and only identifies a high probability of their presence in the southern part of the province of Chachapoyas (district of Leimebamba). The difference in predictions might stem from García et al. [20] not incorporating edaphic variables [22], which can significantly influence the model, considering *Cinchona* species grow in fertile, well-drained soils, mostly with a thick layer of organic matter, a sandy loam texture, and an acidic-to-neutral pH [76,77], often rich in iron, nitrogen, and potassium [78].

Although Amazonas is notable for its high number of conservation areas, this does not necessarily translate into the effective preservation of *Cinchona* species [79]. Approximately 24.29% of the estimated potential distribution for these species in the department is protected within conservation areas (25.51% for *C. capuli*, 21.02% for *C. macrocalyx*, and 26.34% for *C. pubescens*). However, it is estimated that 34.45% of the department's categorized degraded territory overlaps with the potential distribution area of these *Cinchona* species (29.80% for *C. capuli*, 38.72% for *C. macrocalyx*, and 34.82% for *C. pubescens*). The high percentages of degraded areas and the low presence of *Cinchona* species in conservation areas indicate threats from deforestation for agricultural and livestock expansion [16], as well as forest fires that menace the entire genus in Peruvian territory [15]. Consequently, the restoration of this degraded land is feasible through recovery plans that include afforestation and reforestation with *Cinchona* species [80]. The Ministry of Agriculture and Irrigation approved the "Action Plan for Forest Repopulation with *Cinchona* genus species (Quinine Tree) 2020–2022," aiming to produce and plant 145 hectares with *Cinchona* species (*Cinchona calisaya* Wedd., *C. officinalis* L., and *C. pubescens* Vahl) in Amazonas, Cajamarca, Cusco, Huánuco, Junín, Lambayeque, Lima, Pasco, Piura, and Puno [81]. However, the species selection criteria and repopulation zones remain undisclosed [82]. In Amazonas, several public institutions are executing forestry projects, mainly promoting the planting of introduced species like *Pinus* sp., *Eucalyptus* sp., and *Cupressus* sp., among others [83], which often overlooks the native flora species in Amazonas, such as *C. capuli*, categorized as Near Threatened (NT); *C. pubescens*, evaluated as Least Concern (LC) with an unknown population; and *C. macrocalyx*, categorized as Least Concern (LC) with a stable population [84]. Paradoxically, species that are less protected often face greater threats [85].

Therefore, the creation and implementation of a regional action plan for reforestation using species of the genus *Cinchona* is proposed. This strategy aims to conserve the genus, restore forests, preserve natural ecosystems, and contribute to climate change

mitigation [86–88]. In summary, this research was focused on predicting the possible distribution of three species belonging to the genus *Cinchona* in the department of Amazonas, considering current climatic conditions. Additionally, it also analyzed the extent to which this potential distribution overlaps with conservation areas and degraded areas. However, future research could explore the distribution of these species in the context of climate change, similar to a study by Rojas et al. [32], which examined five timber tree species in the Amazonas, Peru. This would be especially relevant, as identifying areas with stable distribution both presently and in the future could be crucial for the success of any conservation or restoration initiative [73,89]. The limitations encountered in this study include the scarcity of available local climatic information, leading us to work with global data obtained from the WorldClim version 2 database (http://www.worldclim.org/ (accessed on 22 June 2023)), as well as the lack of local information on soil variables and the limited presence of records of species in the department. These limitations on the influencing factors could have impacted the model calibration [90]; however, the integration of various environmental variables and real-time data contributed to making our model more accurate compared to previous studies on the genus [20–22]. Therefore, it is recommended that future studies continue investigating distribution models of these and other *Cinchona* species, considering anthropogenic variables, soil variable values from soil analyses, and actual climatic values from the area obtained from meteorological stations, as this will enhance the prediction of potential distribution within the genus [91]. Nevertheless, the information generated in this study holds significant importance as it provides a theoretical foundation for new distribution studies and broader or more specific research that offers insights for forest management, afforestation, and/or the conservation of the montane forests in the southern and central regions of the Amazonas department, which serve as ideal habitats for these *Cinchona* species.

## 5. Conclusions

The successful modeling of the potential distribution of *C. capuli*, *C. macrocalyx*, and *C. pubescens* under current conditions revealed a 23.09% occurrence probability in the Amazonas department. This includes 17.22% for *C. capuli*, 29.11% for *C. macrocalyx*, and 22.94% for *C. pubescens*, spanning an average area of approximately 10,000 km$^2$. Of this, only 24.29% of the potential distribution area for these species is under protection in conservation areas within the department. Meanwhile, 34.45% of the potential area for these species is in a degraded state. This degradation poses a significant threat to the occurrence areas of *C. capuli*, *C. macrocalyx*, and *C. pubescens* in Amazonas, primarily due to deforestation for agricultural and livestock expansion and recurring forest fires. To safeguard the *Cinchona* species, a range of conservation strategies are proposed. These include creating new protected areas and recovering degraded habitats through reforestation efforts with *C. capuli*, *C. macrocalyx*, and *C. pubescens* plantations. The development and implementation of a regional action plan for forest repopulation with *Cinchona*-genus species is suggested. Such strategies are crucial not only for conserving the genus but also for restoring forests, preserving natural ecosystems, and mitigating climate change.

**Supplementary Materials:** The following supporting information can be downloaded at: https://www.mdpi.com/article/10.3390/f15020321/s1, Figure S1: Field collections records of the *Cinchona* genus: (a) *C. capuli*, (b) *C. pubescens*, (c) *C. macrocalyx*., Figure S2: Jackknife test showing how different environmental variables affect the modelling of the distribution of the three *Cinchona* species in the department of Amazonas, northeastern Peru., Figure S3: ROC curve of the MaxEnt model for the three *Cinchona* species. Table S1: Presence records utilized in modelling the potential distribution of the three species of the genus *Cinchona* in the department of Amazonas, northeastern Peru. Table S2: Calculations of the proportions of influence of environmental variables in the MaxEnt model.

**Author Contributions:** Conceptualization, E.C.-C. and G.M.-M.; data curation, J.M.C.T. and R.S.L.; formal analysis, E.C.-C., G.M.-M. and E.P.M.; funding acquisition, M.O.C. and R.E.C.R.; investigation, E.C.-C., G.M.-M., J.M.C.T., E.P.M., M.O.C., R.E.C.R. and R.S.L.; methodology, E.C.-C., G.M.-M., J.M.C.T. and R.S.L.; project administration, M.O.C. and R.E.C.R.; software, E.C.-C., J.M.C.T. and G.M.-M.; supervision, E.P.M., M.O.C. and R.E.C.R.; writing—original draft preparation, E.C.-C.; writing—review and editing, E.C.-C., G.M.-M., J.M.C.T., E.P.M., A.C.-S., M.O.C., R.S.L. and R.E.C.R. All authors have read and agreed to the published version of the manuscript.

**Funding:** This research was mainly financed by CUI Project 2261386 "Creation of Laboratory Services of Genetic Resources of Biodiversity and Conservation of Wild Species of the National University Toribio Rodriguez de Mendoza, Amazonas Region". Other financing entities of this research are the CUI Project N° 312235 "Creation of a Geomatics and Remote Sensing Laboratory Service of the National University Toribio Rodriguez, Amazonas Region"- GEOMATICA, and the Vice Rectorate of Research of the National University Toribio Rodriguez de Mendoza of Amazonas".

**Data Availability Statement:** The data presented in this study are available on request from the corresponding author (privacy).

**Acknowledgments:** The authors are grateful for the support provided by the Instituto de Investigaciones para el Desarrollo Sostenible de Ceja de Selva of the Universidad Nacional Toribio Rodríguez de Mendoza de Amazonas and the KUELAP herbarium for allowing the review of the *Cinchona* botanical samples deposited in their facilities.

**Conflicts of Interest:** The authors declare no conflicts of interest.

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
