# Peer review of "Potential Distribution and Identification of Critical Areas for the Preservation and Recovery of Three Species of Cinchona L. (Rubiaceae) in Northeastern Peru"

_forests, doi:10.3390/f15020321_

Round 1

Reviewer 1 Report

Comments and Suggestions for Authors

The idea of this article is appealing, but there are many shortcomings. For example, the results described in the abstract do not correspond to those described in the results, and the descriptions in the materials and methods are not detailed. Unfortunately, this article has been rejected. But some specific comments are given in the following, and it is suggested to revise and submit again.

1. Abstract

30 – 32 Area units and abbreviations are inconsistent throughout the article

34 – 36 There is a problem with the data, which is inconsistent with the value in the result description, please check.

2. Keywords

39 – "husk" whether appropriate?

3. Introduction
78 – 79 Species names are not consistent.

4. Materials and Methods

84 – Is spelling, correct?

115 – 117 Is the species data cleaned? Is there a spatial autocorrelation problem? In addition, the bioclimatic variables are selected in the current condition (average 1970-2000), while the species records data contains data obtained before 1970. Should these data be used?

160Are the radiation variables resampled?

162 Are the units of the variables in Table 1 all correct?

163 – 173 What criteria are used to select the variables used to build the model? Why do some species choose 8 variables and others 9? There are many writing errors, please check.

175 – 205 Figure 2 and Figure 3 show the optimal clustering number of 19 bioclimatic variables? Or the optimal clustering number of 32 variables?

210 – 211 How many iterations does the maximum entropy model take?

240 – 242 Should the species name be in italics?

5. Results

245 – 248 Why 6 bioclimatic variables? Is not the model constructed using the variables selected in Section 2.4?

245 – 258 The abbreviation of the variable name is inconsistent throughout the article, please check.

251 – 258 The importance of variables in the MaxEnt is carried out by the Jackknife method. Isn't Table 2 obtained by the Jackknife method? Are these three variables the most important for predicting the potential distribution of species C. pubescens? Or is it just Bio06?

377 – How was the degraded area obtained?

322 – 341 Are these numbers correct? The abbreviations of conservation areas are inconsistent.

6. Discussion

445 – 458 The abbreviations for MaxEnt need to be consistent and checked throughout?

494 – 495 Check whether the data is correct?

439 – 520 There was some overlap in the Discussion with the Results.

7. References

Check whether the format of the literature is consistent? Are journal names abbreviated? Are the years in the same position?

Comments on the Quality of English Language

Extensive editing of English language required

Author Response

Good afternoon dear reviewer. I am attaching a word file with the answers to the comments and/or suggestions proposed by you to our article "Potential distribution and identification of critical areas for the preservation and recovery of three species of Cinchona L (Rubiaceae) in northeastern Peru".

Elver Coronel Castro
Corresponding author

Reviewer 2 Report

Comments and Suggestions for Authors

Dear authors and editors. I am pleased to review the manuscript "Potential distribution and identification of critical areas for the preservation and recovery of three species of Cinchona L (Rubiaceae) in northeastern Peru". The work is devoted to an interesting topic. The article is relevant. The article corresponds to the subject of the scientific journal.

1. Figure 1. Inset maps must have their own coordinate grids, scale - otherwise, a visualization is created that Peru is located in the coordinates indicated in the large figure.

2. Figure 5,6, 7. Most likely an error in the coordinate grid and map layout. The quality of the drawing is very poor.

3. The quality of the drawings is very low. It is impossible to review them.

4. Section 2.4 should be improved. It is impossible to reproduce your methodology described in this form. In general, section 2 is written very briefly. This is the main drawback of the work.

5. Specify the limitations of the study

6. Indicate the prospects for future research.

Author Response

(The authors gave the same response as above.)

Reviewer 3 Report

Comments and Suggestions for Authors

The article is devoted to the actual problem of tropical mountain forest conservation. The authors calculated the possible potential distribution of three Cinchona L. (Rubiaceae) species in the north-east of the Peruvian Andean zone using the maximum entropy algorithm (MaxEnt). The variables for the distribution models were obtained from various databases: WorldClim, Digital Elevation Model, Soil Grids. The use of modern approaches determines the novelty of the research. The results obtained are important for Cinchona species conservation and sustainable forest management in Peru.

Paper is well written. However following items need to be addressed.

Introduction

-       The purpose and relevant hypothesis for the study are missing. Instead, the authors indicated what results were obtained. It is also worth emphasizing the novelty and importance of the research for the scientific community.

-       L.60 – “(article in preparation)”. It is worth writing reference or removing this information.

-       There are quite a lot of cited references on old publications (not for the last 5 years). It is better to replace them with modern ones.

Materials and Methods

Methodological approaches are described quite fully. However, Figures 2, 3 are of low quality and therefore not informative.

Results

The research results are presented clearly and illustrated with four figures and four tables. Figure 5 are of low quality and therefore not informative.

Discussion

The authors well described the novelty of the research, differences from previous studies, justification for the choice of methods and discussed some of the results.

Conclusions

Conclusions follow from the results and are reasonable.

Author Response

(The authors gave the same response as above.)

Round 2

Reviewer 1 Report

Comments and Suggestions for Authors

 Although some revisions have been made according to the previous suggestions, the revisions are not comprehensive enough (e.g., line 274, line 421). In addition to formatting issues, several issues are not clearly described. 

1. Are species occurrence records cleaned? 

2. The variable selection steps are not clearly described. 

3. How to identify of critical areas for the preservation and recovery of three species? 

4. How are degraded areas in Peru obtained? 

5. Which period does Shapefile for Peru's National System of Protected Natural Areas represent? 

6. Are the periods of degraded areas, restored areas and protected areas matched? 

Author Response

Good evening dear reviewer.
I am attaching the corrections to the suggestions and/or comments made by you, hoping to clarify your doubts. 

Thank you very much.
